# Regularizing Towards Permutation Invariance in Recurrent Models

**Edo Cohen-Karlik**
Tel Aviv University, Israel
edocohen@mail.tau.ac.il

**Avichai Ben David**
Tel Aviv University, Israel
avichaib@mail.tau.ac.il

**Amir Globerson**
Tel Aviv University, Israel
and Google Research
gamir@post.tau.ac.il

## Abstract

In many machine learning problems the output should not depend on the order of the input. Such "permutation invariant" functions have been studied extensively recently. Here we argue that temporal architectures such as RNNs are highly relevant for such problems, despite the inherent dependence of RNNs on order. We show that RNNs can be regularized towards permutation invariance, and that this can result in compact models, as compared to non-recurrent architectures. We implement this idea via a novel form of stochastic regularization.

Existing solutions mostly suggest restricting the learning problem to hypothesis classes which are permutation invariant by design [Zaheer et al., 2017, Lee et al., 2019, Murphy et al., 2018]. Our approach of enforcing permutation invariance via regularization gives rise to models which are *semi permutation invariant* (e.g. invariant to some permutations and not to others). We show that our method outperforms other permutation invariant approaches on synthetic and real world datasets.

## 1 Introduction

In recent years deep learning has shown remarkable performance in a vast range of applications from natural language processing to autonomous vehicles.

One of the most successful models of the current deep-learning Renaissance are convolutional neural nets (CNN) [Krizhevsky et al., 2012], which utilize domain specific properties such as invariance of images to specific spatial transformations. Such inductive bias is common in other domains where it is necessary to learn from limited amounts of data.

In this work we consider the setting where learned functions are such that the order of the inputs does not affect the output value. These problems are commonly referred to as *permutation invariant*, and typical solutions aim to restrict the learned models to functions that are permutation invariant by design. One such work is DeepSets [Zaheer et al., 2017], which provided a characterization of permutation invariant functions. Specifically, given a set of objects $\{\boldsymbol{x}_1, \ldots, \boldsymbol{x}_n\}$ and a permutation invariant function $f(\boldsymbol{x}_1, \ldots, \boldsymbol{x}_n)$ they showed that $f$ can be expressed as follows: there exist two networks $\varphi(\cdot)$ and $\rho(\cdot)$ such that $f$ is a result of applying $\varphi$ to all inputs, sum-aggregating and then applying $\rho$. Namely:

$$f(\boldsymbol{x}_1, \ldots, \boldsymbol{x}_n) = \rho\left(\sum_{i=1}^{n} \varphi(\boldsymbol{x}_i)\right) \tag{1.1}$$

Other works suggest replacing summation with different permutation invariant aggregation methods such as element-wise maximum [Qi et al., 2017] and attention mechanisms [Lee et al., 2019, Vinyals et al., 2016]. Although these approaches result in permutation invariant functions and can be shown to express any permutation invariant function, it is not clear how many parameters are required for such an implementation. Indeed, there remains an important open question: given the many ways in which a given permutation-invariant function can be implemented, what are the relative advantages of each approach?

Here we highlight the potential of recurrent architectures for modeling permutation invariance. By recurrent architectures we mean any architecture that has a state that evolves as the sequence is processed. For example, recurrent neural networks (RNNs), LSTMS [Hochreiter and Schmidhuber, 1997] and GRUs [Chung et al., 2014]. We focus on standard RNNs in what follows, but our approach applies to any recurrent model. It initially seems counter-intuitive that RNNs should be useful for modeling permutation invariant functions. However, as we show in Section 3 there are permutation invariant functions that RNNs can model with far fewer parameters than DeepSets.

The reason why RNNs are effective models for permutation invariant functions is that their state can be used as an aggregator to perform order invariant summaries. For example, max-aggregation for positive numbers can be implemented via the simple state update $s_{t+1} = \max[s_t, x_t]$ and $s_0 = 0$, and can be realized with only four ReLU gates. Similarly, the state can collect statistics of the sequence in a permutation invariant manner (e.g., order-statistics [Alon et al., 1999] etc.).

One option to achieve permutation invariant RNNs is to take all permutations of the input sequence, feed each of them to an RNN, and average. This method is exponential in the sequence length, and Murphy et al. [2018] suggest approximating it by sub-sampling the permutation set. Here we take an alternative approach that is conceptually simple, more general and more empirically effective. We propose to *regularize* RNNs towards invariance. Namely learn a regular RNN model $f$, but with a regularization term $R(f)$ that penalizes models $f$ that violate invariance. The naive implementation of this idea would be to require that all permutations of the training data result in the same output. However, we go beyond this, by requiring same-output for subsets of training sequences. We call this "subset invariance regularization" (SIRE). It is a very natural criterion since most sequence classification tasks do not have fixed length inputs, and a subsequence of a training point is likely to be a valid example as well.

In contrast to previous methods which all result in architectures that are invariant by design, our method enforces invariance in a "soft" manner, thus enabling usage in "semi" permutation invariant settings where previous methods are not applicable. This makes it applicable to settings where there is *some* temporal structure in the inputs.

The rest of the paper is structured as follows: in Section 2 we define notations and formally describe the problem setting. Section 3 shows that in some cases RNNs are favorable with respect to other permutation invariant architectures. In Section 4 we describe our regularization method. In Section 5 we discuss related work, and Section 6 provides an empirical evaluation.

## 2 Formulation

Consider a general recurrent neural network, with a state update function $f : \mathcal{S} \times \mathcal{X} \to \mathcal{S}$ parameterized by $\Theta$ (we omit $\Theta$ dependence when clear from context). The initial state $\mathbf{s_0}$ is also a learned parameter. The state update rule is therefore given by:

$$\mathbf{s_{t+1}} = f(\mathbf{s_t}, \mathbf{x_{t+1}}; \Theta) \tag{2.1}$$

In what follows we use the notation $f(\mathbf{s}, \mathbf{x_1}, \ldots, \mathbf{x_n})$ to denote the state that is generated by starting at state $\mathbf{s}$ and processing the sequence $\mathbf{x_1}, \ldots, \mathbf{x_n}$. Thus, for an input sequence $(\mathbf{x_1}, \mathbf{x_2}, \mathbf{x_3})$, the state $\mathbf{s_3}$ will be given by:

$$\mathbf{s_3} = f(\mathbf{s_2}, \mathbf{x_3}) = f(f(\mathbf{s_1}, \mathbf{x_2}), \mathbf{x_3}) = f(f(f(\mathbf{s_0}, \mathbf{x_1}), \mathbf{x_2}), \mathbf{x_3}) := f(\mathbf{s_0}, \mathbf{x_1}, \mathbf{x_2}, \mathbf{x_3}) \tag{2.2}$$

The state is mapped to an output $\mathbf{y_t}$ via the output mapping $\mathbf{y_t} = g(\mathbf{s_t})$.

We next define several notions of permutation invariance. Informally, a model $f$ is permutation invariant if it provides the same output regardless of the ordering of the input. In the definitions below we assume input is sampled from some distribution $\mathcal{D}$ and require invariance only for inputs

in the support of $\mathcal{D}$, or their subsets. If $\mathcal{D}$ is the true underlying distribution of the data, then clearly this is sufficient since we will never test on examples outside $\mathcal{D}$. When training we will consider the empirical distribution as an approximation to $\mathcal{D}$.

We begin by defining invariance for the sequences sampled from $\mathcal{D}$.

**Definition 1.** *An RNN is called permutation invariant with respect to $\mathcal{D}$ on length n, if for any $(\mathbf{x_1}, \ldots, \mathbf{x_n})$ in the support of $\mathcal{D}$ and any permutation $\pi \in S^n$ we have:*[1]

$$f(\mathbf{s_0}, \mathbf{x_1}, \ldots, \mathbf{x_n}) = f(\mathbf{s_0}, \mathbf{x}_{\pi_1}, \ldots, \mathbf{x}_{\pi_n}) \tag{2.3}$$

We next note that Definition 1 does not imply any constraint on sequences of length $n' < n$. However, a natural requirement from a permutation invariant RNN is to satisfy the same properties for shorter sequences as well. This is captured by the following definition.

**Definition 2.** *An RNN is called subset-permutation invariant with respect to $\mathcal{D}$ on length n, if for any $(\mathbf{x_1}, \ldots, \mathbf{x_n})$ in the support of $\mathcal{D}$, any sequence $(\mathbf{x'_1}, \ldots, \mathbf{x'_m})$ whose elements are a subset of $(\mathbf{x_1}, \ldots, \mathbf{x_n})$, and any permutations $\hat{\pi}, \tilde{\pi} \in S^m$ it holds that:*

$$f(\mathbf{s_0}, \mathbf{x'}_{\hat{\pi}_1}, \ldots, \mathbf{x'}_{\hat{\pi}_m}) = f(\mathbf{s_0}, \mathbf{x'}_{\tilde{\pi}_1}, \ldots, \mathbf{x'}_{\tilde{\pi}_m}) \tag{2.4}$$

Note that Definition 2 is more restrictive than Definition 1. In particular, an RNN which satisfies Definition 2 also trivially satisfies Definition 1 but the other way around is not true.

Definition 2 involves the response of RNNs to sub-sequences of the data. It is thus closely related to the states that the RNN can reach when presented with sequences of different length. The next definition captures this notion.

**Definition 3.** *Denote the states reachable by $\mathcal{D}$ and parameters $\Theta$ by $\mathcal{S}_{\mathcal{D},\Theta}$. Formally:*[2]

$$\mathcal{S}_{\mathcal{D},\Theta} \overset{\text{def}}{=} \left\{ f(\mathbf{s_0}, \mathbf{x'_1}, \ldots, \mathbf{x'_i}; \Theta) \mid \exists (\mathbf{x}_1, \ldots, \mathbf{x}_n) : (\mathbf{x'}_1, \ldots, \mathbf{x'}_i) \in 2^{\{\mathbf{x}_1, \ldots, \mathbf{x}_n\}}, \, P(\mathbf{x_1}, \ldots, \mathbf{x_n}) > 0 \right\}$$

We shall use this definition when proposing an invariance regularization in Section 4.

## 3   Compact RNNs for Permutation Invariance

In this section we show the existence of functions that are permutation invariant and are modeled by a very small RNN, whereas modeling them with a DeepSet architecture requires significantly more parameters. In what follows we make this argument formal.

**Theorem 4.** *For any natural number $K > 4$, there exists a permutation invariant function that can be implemented by an RNN with 3 hidden neurons but its DeepSets implementation requires $\Omega(K)$ neurons to implement.*

The above theorem says there are cases where an RNN requires far fewer parameters to implement than a DeepSet architecture, and this will of course imply (by standard sample complexity lower bounds) that there are distributions for which the RNN will require far fewer samples to learn than DeepSets. We next prove the result by using the parity function to demonstrate the gap in model size. In Section 6 we provide an empirical demonstration of the result.

*Proof.* In order to prove Theorem 4 one needs to show that for any $K$ there exists a function $f$ such that: (a) $f$ can be implemented with a constant number of neurons using RNNs, and (b) any DeepSets architecture will require at least $K$ neurons to implement $f$.

Let $n = 2^K$. Given $\mathcal{X} = \{0, 1\}$, define the *parity* function operating over sets of size $n$:

$$parity(\{\boldsymbol{x}_1, \ldots, \boldsymbol{x}_n\}) = \left( \sum_{i=1}^{n} x_i \right) mod\ 2 \tag{3.1}$$

Next we claim that the *parity* function can be implemented by an RNN with three hidden neurons and 12 parameters in total. Consider an RNN operating on a sequence $x_1, \ldots, x_n \in \{0, 1\}$ with the

following update rule $s_{t+1} = W_1^T \sigma(W_2 x_t + W_3 s_t + B)$. By setting: $W_1 = [1, -1, -1]$, $W_2 = W_3 = [2, 2, 2]$, and $B = [0, -1, -3]$ we have,

$$s_{t+1} = \begin{pmatrix} 1 \\ -1 \\ -1 \end{pmatrix}^T \sigma \left( \begin{pmatrix} 2x_t + 2s_t \\ 2x_t + 2s_t - 1 \\ 2x_t + 2s_t - 3 \end{pmatrix} \right) \tag{3.2}$$

The above implements the function $ReLU(a) - ReLU(a-1) - ReLU(a-3)$, where $a = 2x_t + 2s_t$. This in turn is equivalent to the *XOR* function (namely $(x_t + s_t) mod\ 2$).

Since the RNN state update implements addition modulo 2, it easily follows that the full RNN will calculate parity. Specifically, by setting $s_0 = 0$ and applying $f(f(s_0, x_1), \ldots, x_n)$ we obtain the parity of $(x_1, \ldots, x_n)$, as required. Note that the implementation requires 4 weight matrices, $W_1, W_2, W_3, B \in \mathbb{R}^3$ which amounts to 12 parameters.

The second part of the proof requires showing that any DeepSets architecture needs at least $K$ neurons to implement *parity* over sets of $n$ elements. Recall that a DeepSets architecture is composed of two functions, $\varphi$ and $\rho$ (see Eq. 1.1). We assume $\varphi$ and $\rho$ are feed-forward nets with ReLU activations and $l$ hidden layers of fixed width $d$. Our result can be extended to variable width networks.

First we argue that WLOG the function $\varphi$ can be assumed to be the identity $\varphi_I(x) = x$. To see this, recall that there are only two possible $x$ values. We will now take any DeepSet implementation $\varphi, \rho$ and show that it has an equivalent implementation $\varphi_I, \tilde{\rho}$. Denote the two values that $\varphi$ takes by $\varphi(0) = \boldsymbol{v}_0$ and $\varphi(1) = \boldsymbol{v}_1$. Thus, after the sum aggregation of the DeepSet architecture we have:

$$\sum_i \varphi(x_i) = n_0 \boldsymbol{v}_0 + n_1 \boldsymbol{v}_1 = (n - n_1) \boldsymbol{v}_0 + n_1 \boldsymbol{v}_1 = n \boldsymbol{v}_0 + n_1 (\boldsymbol{v}_1 - \boldsymbol{v}_0) \tag{3.3}$$

where $n_0, n_1$ are the number of zeros and one in the sequence $x_1, \ldots, x_n$ (so that $n_0 + n_1 = n$). Now note that: $\sum_i \varphi_I(x_i) = \sum_i x_i = n_1$. Then we can define $\tilde{\rho}(z) = \rho(n \boldsymbol{v}_0 + z(\boldsymbol{v}_1 - \boldsymbol{v}_0))$, and

we have that the implementations are equivalent, namely:

$$\rho \left( \sum_i \varphi(x_i) \right) = \tilde{\rho} \left( \sum_i \varphi_I(x_i) \right) \tag{3.4}$$

From now on, we therefore assume $\varphi(x) = x$.

Given the above, we can assume that $\varphi(x) = x$. Therefore $\rho : \mathbb{R} \to \mathbb{R}$ is a continuous function which takes $\sum_{i=1}^n x_i$ as input and outputs 1 for odd values and 0 for even values. Recall that a ReLU network implements a piecewise linear function, and a network of depth $L$ and $r$ units per layer can model a function with at most $r^L$ linear segments [Montufar et al., 2014]. The $\rho$ function above must have at least $n$ segments since it switches between 0 and 1 values $n$ times. This network has $Lr^2$ parameters. Minimizing $Lr^2$ under the condition that $r^L = n$ the minimum is $L = \log_2 n, r = 2$. Thus the minimum number of units in a network that implements $\rho$ is $4 \log_2 n = 4K$, proving the result. $\quad\square$

# 4 Permutation Invariant Regularization

In the previous section we showed that RNNs can implement certain permutation invariant functions in a compact manner. On the other hand, if we learn an RNN from data, we have no guarantee that it will be permutation invariant. The question is then: how can we learn RNNs that correspond to permutation invariant functions. In this section we present an approach to this problem, which relies on a regularization term that "encourages" permutation invariant RNNs. Intuitively, such a term should be designed such that it is minimized only by permutation invariant RNNs.

## 4.1 Regularizing Towards Subset Permutation Invariance

Our goal is to define a function $R(f)$ that will be zero when $f$ is subset permutation invariant and non-zero otherwise.

Following Definition 2 it is natural to define the expected squared error between the RNN state for all sub-sequence pairs that are required to have the same output. Clearly, having the same state will

result in the same output. Thus we define the regularizer:

$$R_{SUB}(f) = \mathbb{E}\left[\left(f(\mathbf{s_0}, \mathbf{x}'_{\hat{\pi}_1}, \ldots, \mathbf{x}'_{\hat{\pi}_m}) - f(\mathbf{s_0}, \mathbf{x}'_{\tilde{\pi}_1}, \ldots, \mathbf{x}'_{\tilde{\pi}_m})\right)^2\right] \tag{4.1}$$

where the expectation is taken with respect to $\mathcal{D}$, $\hat{\pi}, \tilde{\pi} \in S^m$ and the subsequence sampling.

## 4.2 Pair Permutation Invariance Regularization

Calculating the $R_{SUB}$ regularizer exactly will take exponential time, and thus we must resort to approximations. The simplest approach would be to randomly select a subset and two permutations and then replace the expectation in SUB with its empirical average. However, as we show next, there is a simpler approach to regularization.

We next suggest an alternative regularizer, $R_{SIRE}$ that also vanishes for subset-permutation-invariant models, but avoids the permutation sampling in $R_{SUB}$. Our key insight is that because of the recurrent nature of the RNN, one only needs to verify invariance by considering invariance to adding two elements to an existing sub-sequence. We next state the key result that facilitates the new regularizer.

**Theorem 5.** *An RNN is subset-permutation invariant with respect to $\mathcal{D}$ if $\forall \mathbf{x_1}, \mathbf{x_2} \in \mathcal{X}$ and $\forall \mathbf{s} \in \mathcal{S}_{\mathcal{D},\Theta}$ it holds that:*

$$f(\mathbf{s}, \mathbf{x_1}, \mathbf{x_2}) = f(\mathbf{s}, \mathbf{x_2}, \mathbf{x_1}) \tag{4.2}$$

In order to prove Theorem 5, we make use of the following lemma.

**Lemma 6.** *Assume Equation 4.2 holds under the conditions of Theorem 5. Then $\forall \mathbf{x_1}, \ldots, \mathbf{x_t} \in \mathcal{X}$ the following holds for all $i \in \{1, \ldots, t-1\}$:*

$$f(\mathbf{s_0}, \mathbf{x_1}, \ldots, \mathbf{x_{i-1}}, \mathbf{x_i}, \mathbf{x_{i+1}}, \mathbf{x_{i+2}}, \ldots, \mathbf{x_t}) = f(\mathbf{s_0}, \mathbf{x_1}, \ldots, \mathbf{x_{i-1}}, \mathbf{x_{i+1}}, \mathbf{x_i}, \mathbf{x_{i+2}}, \ldots, \mathbf{x_t})$$

**Corollary 7.** *Assume the condition of Lemma 6 holds. Then for a sequence $\mathbf{x_1}, \ldots, \mathbf{x_t}$ any two elements $\mathbf{x_i}$ and $\mathbf{x_j}$ can be swapped without changing the value of the resulting state.*

The proofs for Lemma 6 and Corollary 7 are provided in the Appendix.

*Proof of Theorem 5.* Given an arbitrary permutation $\pi \in S^t$ it suffices to show:

$$f(\mathbf{s_0}, \mathbf{x_1}, \ldots, \mathbf{x_t}) = f(\mathbf{s_0}, \mathbf{x_{\pi_1}}, \ldots, \mathbf{x_{\pi_t}}) \tag{4.3}$$

Denote by $i$ the first index for which $i \neq \pi_i$. Then $\exists j > i$ such that $\pi_j = i$. From Corollary 7 we can sawp $\mathbf{x_{\pi_i}}$ and $\mathbf{x_{\pi_j}}$. This process is repeated until Equation 4.3 is satisfied. $\square$

### 4.2.1 The SIRE Regularizer

The result above implies that testing for subset-permutation-invariance is equivalent to testing the effect of adding two inputs to an existing state. This immediately suggests a regularizer that will vanish if and only if the RNN $f$ is subset-permutation-invariant. We refer to this as the Subset-Invariant-Regularizer (SIRE), and define it as follows:

$$R_{SIRE}(f) = \mathop{\mathbb{E}}_{\substack{\mathbf{x_1}, \mathbf{x_2} \sim \mathcal{D} \\ \mathbf{s} \sim \mathcal{S}_{\mathcal{D},\Theta}}}\left[\left(f(\mathbf{s}, \mathbf{x_1}, \mathbf{x_2}) - f(\mathbf{s}, \mathbf{x_2}, \mathbf{x_1})\right)^2\right] \tag{4.4}$$

The key advantage of SIRE over SUB is that SIRE requires sampling sub-sequences but not permutations. Empirically, we show this translates to much faster learning when using $R_{SIRE}$ (see Appendix).

In practice, we of course do not sum over all states $\mathbf{s} \in \mathcal{S}_{\mathcal{D},\Theta}$ as that will require all permutations over the training data. Instead we randomly sample subsets of training sequences and estimate SIRE via an average over those.

In summary, we propose learning an RNN by minimizing the regular training loss (e.g., squared error for regression or cross-entropy for classification) plus the regularization term $R_{SIRE}(f)$ in Equation 4.4, where it is estimated via sampling. As with any regularization scheme, $R_{SIRE}(f)$ may be multiplied by a regularization coefficient $\lambda$.

# 5   Related Work

In recent years, the question of invariances and network architecture has attracted considerable attention, and in particular for various forms of permutation invariances. Several works have focused on characterizing architectures that are "by–design" permutation invariant [Zaheer et al., 2017, Vinyals et al., 2016, Qi et al., 2017, Hartford et al., 2018, Lee et al., 2019, Zellers et al., 2018].

While the above works address invariance for sets, there has also been work on invariance of computations on graphs [Maron et al., 2019, Herzig et al., 2018]. In these, the focus is on problems that take a graph as input, and the goal is for the output to be invariant to all equivalent representations of the graph.

The most relevant line of work relating to ours is Murphy et al. [2018] which suggests viewing a permutation invariant function as an average of the output of all possible orderings applied to a permutation variant function. As this approach is intractable, the authors suggest a few efficient approximations.

Our work is conceptually different from the above works. These approaches are invariant "by design", either explicitly by implementing a permutation invariant pooling operator [Zaheer et al., 2017] or by approximating such a pooling layer [Murphy et al., 2018]. We take a different approach where we do not attempt to obtain a network which is strictly permutation invariant. Instead, we control the variance-invariance spectrum via a regularization term.

# 6   Experiments

In order to evaluate the empirical effectiveness of our regularization scheme we compare it to other methods for learning permutation invariant models. Finally, we also demonstrate how our regularization scheme is effective in "semi" permutation invariant settings.

**Baselines:** We compare our method (namely SIRE regularization) to two permutation invariance learning methods: DeepSets [Zaheer et al., 2017] and the $\pi - SGD$ algorithm from the Janossy Pooling paper [Murphy et al., 2018]. We used the code provided by [Murphy et al., 2018] for experiments over digits and the code by Lee et al. [2019] for the point cloud experiment. Cross-validation was used for learning all architectures.

## 6.1   Learning Parity

In Theorem 4 we showed that RNNs can implement the parity function more efficiently than DeepSets. Here we provide an empirical demonstration of this fact. As training data we take Boolean sequences $\mathbf{x}$ of length at most ten, where the label is their parity. We train both RNNs and DeepSets on those, and test on sequence length up to 100. Figure 1 show the results, and it can be seen that RNNs indeed learn the correct parity function, whereas DeepSets do not. For both networks we used the minimal width required to perfectly fit the training data. For full details see the Appendix.

## 6.2   Arithmetic Tasks on Sequences of Integers

To evaluate our regularization approach, we consider three tasks used in Murphy et al. [2018].[3] In all tasks the input is a sequence $(x_1, \ldots, x_n)$ where $x_i \in \{0, \ldots, 99\}$. The tasks are: (1) ***sum***: The label is the sum of all elements in the sequence. (2) ***range***: The label is the difference between the maximum and minimum elements in the sequence. (3) ***variance***: The label is the empirical variance of the sequence: $\frac{1}{n} \sum_{j=1}^{n} (x_j - \frac{1}{n} \sum_{i=1}^{n} x_i)^2$.

In Murphy et al. [2018] all tasks were evaluated with $n = 5$, here we perform all experiments with $n = 5, 10, \ldots, 30$.

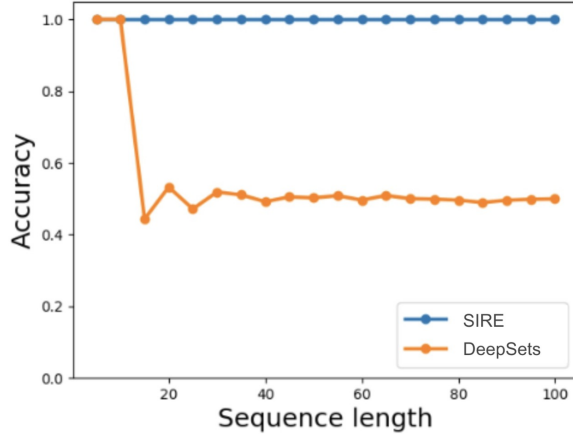

Figure 1: Test accuracy as a function of sequence length for learning parity, using DeepSets and RNNs.

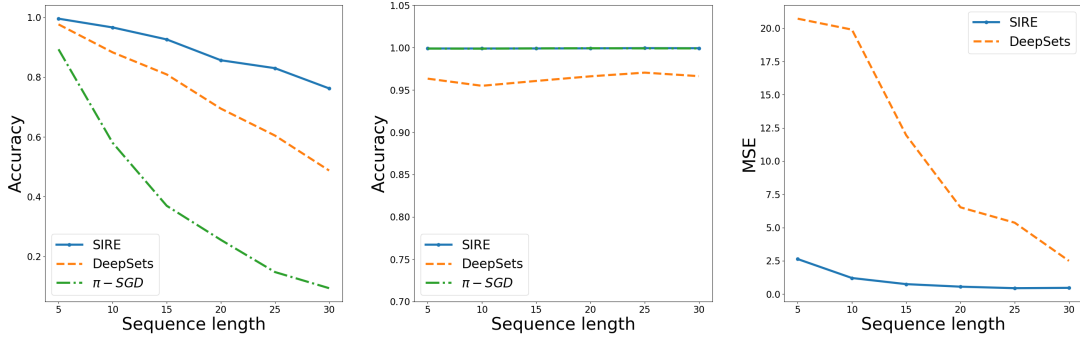

Figure 2: Test prediction accuracy (zero-one error) of *sum* (left) and *range* (center). For the *variance* experiment we report mean square error (as in Murphy et al. [2018]).

Figure 2 shows the average accuracy of each model as a function of the sequence length.[4] The *range* task turns out to be less challenging than the *sum* task. This is probably because the ground truth for a sequence of any size is bounded by 99 (since $\max x_i \leqslant 99$ and $\min x_i \geqslant 0$). In contrast, in the *sum* task, the output is bounded by $99 \cdot n$ which makes the task harder for longer sequences. Results on the *sum* task clearly show that $R_{SIRE}$ generalizes better to longer sequences than the baselines. For the *variance* task we report RMSE values,[5] the graph shows that SIRE outperforms DeepSets. We omit $\pi-$SGD as a baseline in the *variance* task as it failed to converge to low training error with all configurations explored, resulting in poor performance.

## 6.3 Point Clouds

We evaluate our method on a 40-way point cloud classification task using ModelNet40 [Chang et al., 2015]. A point-cloud [Chang et al., 2015, Wu et al., 2015] consists of a set of $n$ vectors in $\mathbb{R}^3$ and has many applications in the growing trend of robotics where LIDAR sensors are common. As point-clouds are represented using a list of vectors, which do not induce a natural order, they are ideal candidates for evaluation of permutation invariant methods. Experiment details are as in [Zaheer et al., 2017].

| Method | 100 pts | 1000 pts | 5000 pts |
|---|---|---|---|
| DeepSets | 0.825 | 0.872 | **0.90** |
| SIRE | **0.835** | **0.878** | 0.899 |

Table 1: Point cloud classification results.

In Table 1 we report results for $n = 100, 1000, 5000$. For $n = 100, 1000$ SIRE outperforms DeepSets and achieves comparable results for $n = 5000$.[6]

## 6.4 Arithmetic Semi Permutation Task

One advantage of our method is the possibility to tune the level of invariance an RNN should capture. This may be useful in real-world datasets where the data is permutation invariant to some extent. For example, *human activity recognition* signals often correspond to repetitive action such as walking, running, etc. Another example is classification of ECG readings which are also characterized by periodic signals.

| | seq. len=10 | seq. len=15 | seq. len=20 |
|---|---|---|---|
| $\lambda = 0.0$ | 0.9346 (0.006) | 0.9461 (0.001) | 0.9678 (0.005) |
| $\lambda = 0.01$ | **0.9584 (0.008)** | **0.9658 (0.008)** | **0.9780 (0.004)** |

Table 2: Learning Semi Permutation Invariant models on the *half-range* problem. Test accuracy for two regularization coefficients and different sequence length. Note that setting $\lambda = 0$ amounts to vanilla RNN without regularization.

Here we demonstrate that our regularization method can capture such "soft" permutation invariance by defining the toy task of *half-range*. The data is a sequence of integers generated in a similar fashion to the *range* task above, but not in a completely invariant manner. The target in this task is to predict the difference of the maximum integer from the first half of the sequence and the minimum integer from the second half of the sequence. Formally, given $(x_1, \ldots, x_k)$, *half-range* is defined as

$$HR(x_1, \ldots, x_k) = \max\left\{x_1, \ldots, x_{\lfloor \frac{k}{2} \rfloor}\right\} - \min\left\{x_{\lfloor \frac{k}{2} \rfloor + 1}, \ldots, x_k\right\}$$

Clearly *half-range* is not permutation invariant. Despite not being completely permutation invariant, the output of *half-range* is not sensitive to many of the possible permutations. Thus, it makes sense to learn it by regularizing towards invariance using the SIRE regularizer. Result for regularized and un-regularized models are shown in Table 2. It can be seen that the regularized version consistently outperforms the standard RNN, showing that the notion of semi-invariance is empirically effective in this case.

## 6.5 Locally Perturbed MNIST

Since the introduction of the MNIST dataset [LeCun et al., 1998], it was used as a starting point for many variations. For example, [Larochelle et al., 2007] created Rotated-MNIST, a more challenging version of MNIST where digits are rotated by a random angle. Another example is MNIST-C [Mu and Gilmer, 2019], where a corrupted test set was created to evaluate out-of-distribution robustness of computer vision models. Yet another variant of MNIST is Perturbed MNIST [Goodfellow et al., 2013, Srivastava et al., 2013], where random permutations are applied to digits.

Here we present *Locally Perturbed MNIST*, a variant of MNIST where pixels are randomly permuted with nearby pixels (see Figure 3). Our goal is to test the performance of our method on data that exhibits some degree of permutation invariance. We believe that such structure is also present in problems such as activity recognition and document analysis.

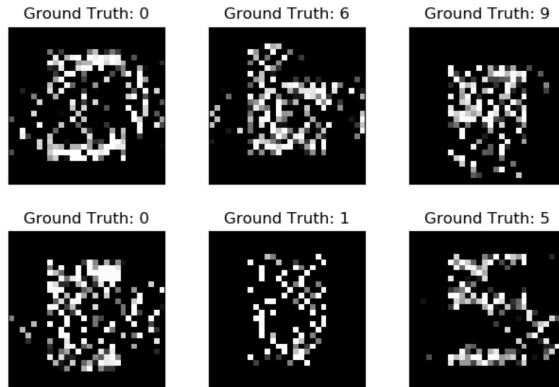

Figure 3: Locally Perturbed MNIST digits. See Appendix for complete details on the perturbation applied.

Since the spatial structure of the image is partially preserved, a relevant baseline is a CNN.[7] The CNN achieves accuracy of $0.963$ on Locally Perturbed MNIST. Learning with a GRU achieves accuracy of $0.833$. Adding $R_{SIRE}$ to the same architecture boosts performance to **0.977**. These findings suggest that $R_{SIRE}$ is effective for learning data with partial invariance properties.

# 7 Discussion

We have introduced a novel approach for modeling permutation invariant learning problems by using recurrent architectures. While RNNs are generally order dependent, we suggest a regularization term and prove that when this term is zero, the network is permutation invariant. We further discuss the permutation invariant *parity* function, for which fixed aggregation based methods such as DeepSets need a large number of parameters whereas simple recurrent models can implement *parity* with $O(1)$ parameters. Empirically, we show that recurrent models can easily solve tasks where DeepSets models do not do as well. We further demonstrate that our method scales better to larger set sizes compared to the recurrent method of Murphy et al. [2018].

In addition to the above, we consider a setting where the data is partially permutation invariant. This property cannot be captured by architectures that are fully permutation invariant by design, and therefore this non-invariant case is typically solved using RNNs. We show that adding our regularization term helps in such "semi" permutation invariant problems. A common approach to solve time series classification is to perform some sort of feature engineering with a sliding window and then treating the resulting features as a set of features without significance to order. Our method may prove useful in such scenarios.

An interesting theoretical question which we do not discuss is that of optimization issues for different architectures. For example, we noticed empirically that our regularization method leads to better optimization errors than RNN without regularization. We hypothesize that this is because the regularization operates on shorter sequences, and can thus alleviate optimization issues related to vanishing gradients.

We note that our regularizer does not require labeled data, and can thus employ unlabeled data in a semi-supervised setting.

Taken together, our results demonstrate the potential of recurrent architectures for permutation invariant and partially invariant modeling. They also serve to further highlight the importance of sample complexity considerations when learning invariant functions. Namely, different implementations of an invariant function may require a different number of parameters and thus result in different sample complexities. In future work we plan to provide a more comprehensive theoretical account of these phenomena.

## 8 Broader Impact

In this work, we analyze an approach for learning recurrent models in cases where there is an underlying permutation invariance. The method can improve sequence labeling systems. We do not see any ethical aspects with the contribution. Societal aspects are positive in terms of improving accuracy of models in healthcare for example.

## 9 Acknowledgements

This project has received funding from the European Research Council (ERC) under the European Unions Horizon 2020 research and innovation programme (grant ERC HOLI 819080).

## Footnotes

[1]Where $S^n$ denotes the symmetry group containing all permutations of a set with $n$ elements.

[2]We use the notation $2^{\{\mathbf{x}_1, \ldots, \mathbf{x}_n\}}$ to denote all sequences whose elements are subsets of $(\mathbf{x}_1, \ldots, \mathbf{x}_n)$.

[3]The original task includes 2 more tasks which we omit since all models achieved near perfect performance.

[4]For each sequence length we perform cross validation to select the best configuration and report the average of 20 runs for *sum* and the average of 3 runs for *range* and *variance*.

[5]Lower is better.

[6]We also evaluated Set Transformer [Lee et al., 2019], using the official implementation (`https://github.com/juho-lee/set_transformer`). We were not able to reproduce reported results for the Set Transformer model, and thus do not report results for it. In addition we evaluated $\pi$−SGD, but it exhibited optimization difficulties for lengths greater than $n = 100$, resulting in poor results compared to other baselines.

[7]We use a simple CNN with 2 convolution layers followed by 2 fully connected layers. This architecture achieves accuracy of $0.9963$ on regular MNIST.

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
