[Supplementary Material]

# Supplementary Information for "Regularizing Towards Permutation Invariance in Recurrent Models"

**Edo Cohen-Karlik**
Tel Aviv University, Israel
edocohen@mail.tau.ac.il

**Avichai Ben David**
Tel Aviv University, Israel
avichaib@mail.tau.ac.il

**Amir Globerson**
Tel Aviv University, Israel
and Google Research
gamir@post.tau.ac.il

Here we provide proofs for the results in the paper, as well as additional information about experiments, and further evaluations.

## Appendix A

**Missing Proofs**

*Proof of Lemma 6.* Denote

$$\hat{\mathbf{s}} \stackrel{\text{def}}{=} f(\mathbf{s_0}, \mathbf{x_1}, \ldots, \mathbf{x_{i-1}}) \tag{.1}$$

Substituting Equation .1 into $f(\mathbf{s_0}, \mathbf{x_1}, \ldots, \mathbf{x_t})$, we have:

$$f(\hat{\mathbf{s}}, \mathbf{x_i}, \mathbf{x_{i+1}}, \mathbf{x_{i+2}}, \ldots, \mathbf{x_t}) \tag{.2}$$

Equation .2 can be written as (see Section 2)

$$f(f(\hat{\mathbf{s}}, \mathbf{x_i}, \mathbf{x_{i+1}}), \mathbf{x_{i+2}}, \ldots, \mathbf{x_t}) \tag{.3}$$

Using Assumption 4.2, we can write Equation .3 as:

$$f(f(\hat{\mathbf{s}}, \mathbf{x_{i+1}}, \mathbf{x_i}), \mathbf{x_{i+2}}, \ldots, \mathbf{x_t}) \tag{.4}$$

Plugging back the simplified notation of nested applications of $f$ and using the definition of Equation .1, the above yields:

$$f(\mathbf{s_0}, \mathbf{x_1}, \ldots, \mathbf{x_{i-1}}, \mathbf{x_{i+1}}, \mathbf{x_i}, \mathbf{x_{i+2}}, \ldots, \mathbf{x_t}) \tag{.5}$$

which concludes the proof. □

*Proof of Corollary 7.* Assume WLOG $i < j$. We need to show that under the conditions of Lemma 6, the following holds

$$f(\mathbf{s_0}, \mathbf{x_1}, \ldots, \mathbf{x_{i-1}}, {\color{red}\mathbf{x_i}}, \mathbf{x_{i+1}}, \ldots, \mathbf{x_{j-1}}, {\color{red}\mathbf{x_j}}, \mathbf{x_{j+1}}, \ldots, \mathbf{x_t}) = \tag{.6}$$
$$f(\mathbf{s_0}, \mathbf{x_1}, \ldots, \mathbf{x_{i-1}}, {\color{red}\mathbf{x_j}}, \mathbf{x_{i+1}}, \ldots, \mathbf{x_{j-1}}, {\color{red}\mathbf{x_i}}, \mathbf{x_{j+1}}, \ldots, \mathbf{x_t})$$

From Lemma 6 we can replace $\mathbf{x_i}$ and $\mathbf{x_{i+1}}$, which yields:

$$f(\mathbf{s_0}, \mathbf{x_1}, \ldots, \mathbf{x_{i-1}}, \mathbf{x_{i+1}}, {\color{red}\mathbf{x_i}}, \mathbf{x_{i+2}}, \ldots, \mathbf{x_{j-1}}, {\color{red}\mathbf{x_j}}, \mathbf{x_{j+1}}, \ldots, \mathbf{x_t}) \tag{.7}$$

This process can be repeated for $j - i - 1$ times, resulting in:

$$f(\mathbf{s_0}, \mathbf{x_1}, \ldots, \mathbf{x_{i-1}}, \mathbf{x_{i+1}}, , \ldots, \mathbf{x_{j-1}}, {\color{red}\mathbf{x_i}}, {\color{red}\mathbf{x_j}}, \mathbf{x_{j+1}}, \ldots, \mathbf{x_t}) \tag{.8}$$

Similarly, swapping $\mathbf{x_j}$ with the elements preceding it for $j - i$ times will result in the RHS of Equation .6, concluding the proof.

□

## Appendix B

**Parity Experiment Details**

Both networks were trained with 1000 randomly generated binary sequences with lengths between 2 and 10. For the RNN, 20 neurons were sufficient for convergence to zero training error. We use a DeepSet with one hidden layer for the preprocessing network $\varphi$, and one hidden layer for the aggregating network $\rho$. Both the $\varphi$ and $\rho$ have a width of 100 which was the minimal width required for convergence for the architecture used. The test set consists of 3000 examples and was generated in a similar fashion to the train set.

**Arithmetic Tasks on Sequences of Integers**

The range of integers used is $\{0, \ldots, 99\}$ for all experiments. The sum experiment was repeated twenty times, and the others three times. We report average accuracy.

Since the tasks defined are regression tasks in nature, we follow Zaheer et al. [2017], Murphy et al. [2018] and use an $L_1$ loss for training. At test time, we round the output of the network to the closest integer and report accuracy using the zero-one loss. For the variance task we report mean squared error (MSE).

**Point Cloud Experiment**

Implementation of point-cloud experiments was based on the official repository of Set Transformers.[1] We omit Set Tranformer [Lee et al., 2019] from the comparison as it did not reproduce the reported results. For DeepSets and our method we use the same architectures used in the DeepSets experiments [Zaheer et al., 2017]. The preprocessing network $\varphi$ is a feed forward neural net with three hidden layers of width 256 and *TanH* activations. For the output network, $\rho$, we use a similar network with one hidden layer and add dropout with a rate of 0.5.

In order to train SIRE we use a GRU with a single layer with width 256 for $n = 100$ and $n = 1000$, for $n = 5000$ we use a width of 512. We use Adam optimizer with a learning rate of $1e^{-3}$. We apply a dropout rate of 0.75 in the GRU layer and a batch size of $n = 200$. We use a regularization coefficient of 0.1 for all sizes. All hyperparameters were selected using cross validation. For $n = 5000$ we use Truncated Back Propagation Through Time with a window of size 1000.

**Locally Perturbed MNIST**

In order to generate Locally Perturbed MNIST we flatten each digit to a 784 dimensional vector. We then perform a "convolution" like operation with full stride. At each window we apply a random permutation. This process limits the distance of a pixel from its original position by at most the window size. We perform the above process twice with window sizes 4 and 7 (Figure **??**).[2]

## Appendix C

**Comparison of $R_{SUB}$ and $R_{SIRE}$**

In the main text, we considered two possible regularizers: SUB and SIRE. Both had a value of zero for permutation invariant models but are otherwise different. As we argue in the main text, SIRE is expected to perform better under a given budget of samples, since it enumerates over the state space more efficiently. In order to empirically evaluate this effect, we perform the *sum* experiment using 200 training examples with sequence length 10 over $\{0, \ldots, 19\}$. We evaluate three regularization coefficient values, $\lambda = 0.001, 0.01, 0.1$ for each scheme. The best values on holdout are $\lambda_{SIRE} = 0.1$ and $\lambda_{SUB} = 0.001$. Each experiment was repeated 5 times, and Figure 1 shows the results averaged over these runs.

Figure 1: Loss curve of training for 1000 epochs with each regularizer.

It can clearly be seen that using $R_{SIRE}$ results in faster convergence. Furthermore, the test accuracy obtained by $R_{SIRE}$ is **0.792 (0.09)** compared to an accuracy of **0.759 (0.11)** achieved by $R_{SUB}$. Thus, we conclude that in this case SIRE outperforms SUB both in convergence speed and test accuracy of the resulting model.

## Footnotes

[1] https://github.com/juho-lee/set_transformer

[2] Resulting in an offset of at most 11 from the pixels original location.