[Reviews · NeurIPS 2020]

Review 1

Summary and Contributions: This paper proposes to model permutation-invariant functions using recurrent models and introduces a permutation-invariance regularizer. This is in contrast to the traditional approach toward learning set functions, which impose permutation invariance in the model by design, as opposed to using a regularizer. As motivation, this paper proves that there exist functions which are more efficiently represented by recurrent models, which only have to learn a "local aggregation" rather than a global set function. Post Author Feedback: =================== I appreciate the authors' response to my questions, which address my low-level questions. The main weakness of the paper is still the experimental section; in addition to limited tasks, reviewer 4 brings up additional simple baselines such as using alternative pooling operations. I still like the main idea of the paper, and am keeping my score of 6.

Strengths: - The main observation in this paper is quite nice - The paper is well motivated, and the methodology makes sense

Weaknesses: - The experiments are all synthetic and somewhat toy

Correctness: The method is correct

Clarity: The paper is well written and easy to understand

Relation to Prior Work: Previous work is adequately covered

Reproducibility: Yes

Additional Feedback: Typos and notation: - Definition 3 seems to be missing the dependence on $\mathcal{D}$. Is $(x_1, \dots, x_i)$ meant to be a sequence in the support of D? - Above equation (3.4): I think you want to define $\tilde{\rho}(z) = \rho(nv_0 + z(v_1 - v_0))$ - Equation (4.4): I think $x_1, x_2$ are drawn from ${\cal S}$ instead of ${\cal D}$ - Line 252: missing a + in this equation? Additional comments: - Some of the experimental results seem hardly believable and could be elaborated on. In particular, the "sum" task being the hardest task is difficult to interpret. First, why is it harder than "unique sum", even when the sequence length is less than 10 (the number of possible distinct inputs)? Second, how can an MLP based model such as DeepSets fail at this task for n=15, when there are clearly trivial functions that can compute this? - It would be nice to see the results of the other baselines on "half range" just for comparison


Review 2

Summary and Contributions: The authors propose to learn a permutation invariant RNN model through regularization rather than inductive bias, by including a loss term that encourages the RNN output to be invariant to transpositions.

Strengths: The observation that sequential symmetric functions are more efficiently learnable with RNNs than DeepSets is an insightful one, and most of the synthetic experiments seem to confirm the efficacy of the method.

Weaknesses: I'm somewhat concerned about other elements of the experiment section. The DeepSets paper included two baselines for the point cloud experiment, with the larger model performing substantially better (up to about 90% accuracy). It seems the authors use the smaller model in their experiment. This mainly raises the question of if the RNN model would be able to scale up as well as the DeepSets model. As this is the only experiment on real data, it's very hard to judge the proposed method without that evaluation.

Correctness: The claims and methodology appear correct.

Clarity: The paper is mostly well-written with some slightly confusing elements, for example the notation for subsequences in Definition 2 and the lack of a definition for P in Definition 3.

Relation to Prior Work: The relation to other works in symmetric neural networks is clear.

Reproducibility: No

Additional Feedback: Post-Rebuttal: The author response suggests the focus of the paper is the representational result in Theorem 4, but the paper seems to be claiming that the proposed RNN model is competitive with DeepSets. Based on the difficulties pointed out by reviewer 4 with using the RNN model, I think that claim requires a stronger experimental section to demonstrate that the representational difference (1) appears in practice and (2) is significant enough to merit the difficulty of a recurrent model. Therefore, I will keep my score the same.


Review 3

Summary and Contributions: The paper proposes a regularization method to adapt RNNs to the case of sets as input, by making the RNNs more invariant to input permutation. This can serve as a simpler and more parameter-efficient alternative to Deep Sets (Zaheer et al.). The paper illustrates an example where a permutation invariant function can be represented by an RNN with O(1) parameters while Deep Sets representation requires at least Omega(log n) parameters, where n is the sequence/set length. The effectiveness of the proposed regularizer is validated on synthetic tasks (e.g., parity of sequence of 0/1, arithmetic tasks on sequence on integers, point clouds). The proposed method shows similar performance to baselines (DeepSets, Janossy pooling) on short sequences (length 5-10) and beter performance on longer sequences (length 15).

Strengths: 1. The idea of Subset-Invariant-Regularizer (SIRE) is a simple and clever way to ensure invariance. Imposing the stronger condition of subset invariance (instead of just permutation invariance), it leads to a much simpler condition of the RNN being invariant to the order of inputs in a pair. This simpler condition leads to a regularizer that's easier to approximate. 2. The example (Section 3) showing a case where RNN being more parameter efficient than Deep Sets is insightful and illustrative. The parity function is permutation invariant, and can be represented by an RNN with O(1) parameters while Deep Sets representation requires at least Omega(log n) parameters, where n is the sequence length. I enjoyed reading this section. 3. The proposed method (RNN + regularization) outperforms Deep Sets and Janossy pooling on longer sequences (length 15), on synthetic tasks. The proposed method also shows slight improvement in a toy task (half-range) where partial invariance is useful.

Weaknesses: 1. All tasks are toy/synthetic, and there is no experiment result on more realistic tasks. Section 6.4 mentions "human activity recognition" and ECG readings. It would be interesting to see if SIRE helps there. That would make the results stronger. 2. Lack of details/reproducibility. The procedure to approximate SIRE by sampling is not described in details. This might make it hard to reproduce the results. In particular, in the definition of SIRE (eq 4.4), one is to sample s from S_{D, \Theta}. However, S_{D, \Theta} is a set, not a distribution. Should one sample from the uniform distribution on this set (which I think is hard)? Or any distribution would work? How do different distribution affect the result? In practice, how many samples should it be averaged over? Is this tuned with cross-validation, or does the same hyperparam work across experiments?

Correctness: The claims and empirical methodology are correct. I have checked the proofs in the appendix (they are short).

Clarity: The paper is easy to read. Some minor questions/comments on clarity: 1. The motivation for the method (end of page 1): it's not clear what the problems with previous approaches are. Is that they use more parameters than necessary? 2. Definition 3 is not clear. The variables i and t are unbounded. I think this is for any i and any t?

Relation to Prior Work: Differences to prior show are clearly discussed. There is sufficient novelty in the proposed method, to the best of my knowledge.

Reproducibility: No

Additional Feedback: Questions/Suggestions: 1. What is the complexity / runtime penalty of sampling s in estimating SIRE? 2. [Minor] It would be interesting to work out what architecture of RNN (e.g. constraints on weight matrices and nonlinearity) will yield this invariance as defined in eq. 4.4. For example in the case of x and s always being 0, 1. This would give more intuition on the this subset-invariance regularizer. 3. [Minor] In Theorem 5, it shoudl be "if and only if"? 4. [Minor] In section 6.2, SIRE performs well up to sequence of length 15. How far can one push the sequence length until it no longer works? 5. [Minor] In section 6.4, I don't have great intuition what kind of "semi"-invariance is compatible with SIRE. So it's only invariant to *some* permutation, but what kind? Suggestion for additional citation: - Wagstaff, Fuchs, Engelcke, Posner, and Osborne. On the Limitations of Representing Functions on Sets. ICML 2019. They show the function in Deep Sets has to be highly discontinuous, similar in spirit to Theorem 4. ===== Update after authors' rebuttal: Thanks for clarifying the sampling procedure of SIRE. As remarked by other reviewers, the argument in the paper (RNN enjoying representation advantage compared to DeepSets) could be made stronger by empirical results comparing to DeepSets on larger and more realistic datasets.


Review 4

Summary and Contributions: This paper proposes a regularization scheme for RNN to be close to permutation invariant. Unlike the previous works that are permutation invariant by design, this work proposes to regularize RNNs towards permutation invariancy. A permutation invariant promoting regularizer is proposed based on an observation that if an RNN is commutative for a step it would be permutation-invariant for arbitrary length sequences.

Strengths: - This work adds an interesting new direction to the literature regarding the permutation-invariant neural networks / set-input neural networks. - Paper is generally well written with helpful illustrative examples.

Weaknesses: - The main motivation to use RNN for permutation-invariant tasks is not fully justified. See below. - The experiments are not through. See below.

Correctness: The main argument to derive the regularizer seems to be correct.

Clarity: Yes, the paper is well written and easy to follow.

Relation to Prior Work: The paper is doing a good job of reviewing the literature and clearly states the difference from the previous works.

Reproducibility: Yes

Additional Feedback: Although I find the idea of regularizing RNN with the proposed regularizer interesting, I’m still unsure of why we should really consider using RNN instead of existing permutation invariant architectures by design (e.g., Deep Sets, Set Transformers, …). The authors are demonstrating the parameter efficiency of RNN compared to Deep Set architecture using the parity example. This argument is valid only when we constrain Deep Set to use the sum pooling operation. One can actually consider various types of pooling operation, for instance featurewise min, max, or even featurewise XOR operation. Deep Sets with featurewise XOR pooling would be as efficient as (or even with fewer parameters) as RNNs. As for the experiments, as far as I understand from the article the Deep Sets are using only the sum pooling operator. Again, this may not be a fair comparison, for instance the reference [18] considers various pooling operations such as mean/max/min. One can also consider a concatenation of all those pooling operations. One critical downside of RNN, in my opinion, is its sequential nature in computation. Unlike the Deep Sets (implemented with feedforward neural nets) or set transformers that can process elements of sets in parallel, RNN requires sequential computation that cannot be parallelized. This can be problematic for both training and inference for long sequences. For instance, consider the amortized clustering example presented in [18], where the size of sets typically scales to several hundred to thousand. For such data, training should suffer from a typical problem that might happen for an RNN being trained with long sequences, and the inference would take much longer time than the Deep Sets or set transformers. Also, this paper only considers permutation invariancy, but permutation equivariancy is also an import property one might ask for. Is it still straightforward to build a permutation equivariant network using RNN by using the proposed regularizer? In summary, I see no strong reason to consider RNN instead of existing permutation invariant networks for the problems requiring permutation invariancy. It would be good for the authors to present more examples of the problems requiring semi-permutation invariancy, as in section 6.4. =============================================== Post author feedback I thank authors for their clarification. However, their feedback didn't really resolve the issue I raised. In order to claim that the proposed method does better than existing permutation-invariant by design methods without having to carefully design pooling functions, more experimental results are required (unless there is a strong theoretical guarantee). At least, the proposed method should be compared to DeepSets nontrivial pooling functions (max/min/xor, ... or concat of them (which I expect to be quite strong baseline). Otherwise, considering the sequential nature of RNN, I still don't see a strong reason to consider the proposed method as an alternative to existing ones. I keep my score as is.

[Author Response · NeurIPS 2020]

**Review 1**: We thank the reviewer for the comments and suggestions. We will fix all typos pointed out in the review. Below we address the two main issues raised by the reviewer .

**How is "sum" harder than "unique sum"?** Indeed accuracy on the *unique-sum* task is better than those of *sum*. The reason is that for unique-sum the input domain is $\{0, \ldots, 9\}$, and for sum it is $\{0, \ldots, 99\}$ (we followed the setup in the Janossy paper). This makes it possible to learn unique-sum by simply having a 10 dimensional state where dimension $i$ is "on" when digit $i$ appears. For the rebuttal, we trained unique-sum with $\{0, \ldots, 99\}$ and accuracy was much lower than sum.

**How can DeepSets fail for** $n = 15$**?** DeepSets failure here is due to convergence to a bad local minimum. We used the standard optimization setup for DeepSets (in the Janossy code). We believe that one advantage of SIRE is improved robustness to optimization failures, because the regularizer also makes optimization easier by using shorter sequences.

**Baseline results for "half range".** For the rebuttal, we ran the *half-range* experiment with DeepSets and it failed to converge to a good solution, resulting in a model which outputs predictions based on the statistics of the data. The resulting accuracy results numbers are (std in brackets): $0.050(4 \cdot 10^{-6}), 0.077(1.4 \cdot 10^{-4}), 0.0895(2.5 \cdot 10^{-7})$ for $n = 10, 15, 20$. These are much worse than what SIRE obtains.

**Review 2**: We thank the reviewer for the comments. We will clarify definitions 2 and 3 according to the suggestions.

**Scalability of the proposed method.** The main focus of our work is to highlight the representational advantages of RNNs, which makes them a very relevant part of the toolbox in invariant modeling. While this comes at a certain computational cost, we believe there are many cases where the benefit in accuracy will make this attractive in practice as demonstrated theoretically and empirically throughout the paper. A similar trade-off is evident in natural language processing where recurrent methods such as GPT are widely used though there exists competing non-recurrent language models such as BERT.

**Review 3**: We thank the reviewer for the through review and insights. We address each point raised below.

**How is SIRE sampled?** We estimate $R_{SIRE}$ by applying a random permutation over input sequences $x_1, \ldots, x_n$ followed by truncating the permuted input sequence to result in two sequences: $x'_1, \ldots, x'_{k-1}, x'_k$ and $x'_1, \ldots, x'_k, x'_{k-1}$ (i.e., the two last elements are switched). This simple scheme worked well, but we agree that other distributions can be explored. Also, note that when the data is generated from $S_n$ the above scheme is uniform over $\mathcal{S}_{\mathcal{D}, \Theta}$.

**What is the complexity of sampling s in estimating SIRE?** As mentioned above, SIRE is estimated by sampling sequence pairs. The sampling operation is linear in sequence length, and then each one of the RNN is applied to the two sequences. Thus, cost is similar to regular training.

**Number of SIRE Samples.** For each original input sequence, we sample just one pair of sequences for the SIRE regularizer. Of course, for a given sequence, different pairs may be sampled each time this sequence is used.

**Def. 3, variables i and t.** Indeed, the definition requires $2 \leq t \in \mathbb{N}$ and $i < t$. We will clarify this.

**In Theorem 5, it should be "if and only if"?** The condition in the theorem is slightly stronger than subset-permutation-invariance, because the sequence that results in adding $x_1, x_2$ may not correspond to a permutation of an actual input sequence. The theorem can be easily adapted to make it "iff", but notation would be more cumbersome.

**What kind of "semi"-invariance is compatible with SIRE?** In problems that are not permutation invariance, SIRE will seek RNNs that are "as close as possible" to permutation invariant, where closeness is measured by SIRE. It will indeed be interesting to further analyze this tradeoff, and we leave this for future work.

**How far can one push the sequence length until it no longer works?** The answer to this question depends heavily on the task at hand and is mostly a matter of optimization. For the sum task, performance begins to deteriorate on relatively short sequences - for $n = 15$ the average accuracy of an RNN using SIRE regularization is 0.957 (0.022). For the purpose of the rebuttal we ran the same experiment with $n = 17$ and got an accuracy of 0.913 (0.003).

**Review 4**: The authors wish to thank the reviewer for the comments and points raised. We address the concerns below.

**Can the proposed method be used for permutation equivariance?** Yes. In a multi-output setting, we can add a regularizer similar to SIRE, but that requires that adding $x_1, x_2$ to state $s$ will generate outputs $y_1, y_2$ whereas adding $x_2, x_1$ will generate outputs $y_2, y_1$. This will result in similar guarantees to what we have, but for equivariance.

**Alternative aggregation methods.** This is indeed a main motivation for our work. In DeepSets and related methods, the choice of aggregation method is key to obtaining good performance. While in principle one can check $\{min, max, sum\}$ there are many other possibilities. On the other hand, in our case, the aggregation function is learned automatically by the RNN, and as we show, the RNN works for cases corresponding to different aggregations.

**Sequential computation.** Recurrent architectures are used extensively in many machine learning applications, from generative models in NLP (e.g., the GPT model is state of the art in language generation, and arguably outperforms BERT which is non-recurrent) to video processing (where 3D convolutions and RNN over 2D convolutions are both commonly used). The reviewer is correct that their sequential nature makes them less amenable to GPU speedups. However, their representational power compensates for this limitation in many cases. Our argument in this paper is that RNNs are very relevant for permutation invariant modeling, and we demonstrate this empirically in several cases.

[Meta-Review · NeurIPS 2020]

Regularizing toward permutation invariance is a simple alternative to explicitly building such an invariance into a model. The reviewers raised some good points, which should be addressed in the camera ready version of the paper. Importantly, the main argument of the paper hinges on the assumption that DeepSets uses a particular reduction operator (namely, \sum), whereas the parity example could be trivially achieved with an exclusive-or reduction. Put differently, circuit complexity arguments are obviously tied to the architectural assumptions, and this paper uses somewhat of a strawman argument to claim RNNs are more efficient. On the other hand, I disagree with the reviewer's argument r/e parallelization of DeepSets, as there is still at least a logarithmic complexity in the reduction. (It would be interesting to explore whether the RNNs designed in this paper could be modified to perform an associative scan rather than a linear scan, and thereby achieve logarithmic complexity.). The reviewers also questioned the limited experimental results. I think this is a significant concern, and I encourage the authors to address it in the camera ready, but it is not one that necessarily precludes publication in my opinion. Finally, I would like to see more discussion/investigation of if and where the regularized models fail to achieve permutation invariance; for large set functions you would need exponentially many samples to assure that all invariances are checked, and presumably this rarely is achieved in practice. Do the learned models still show permutation invariance even on examples they never saw in training?